# Climatically sensitive transfer of iron to maritime Antarctic ecosystems by surface runoff

Andy Hodson[1,2], Aga Nowak[1], Marie Sabacka[3], Anne Jungblut[4], Francisco Navarro[5], David Pearce[6], María Luisa Ávila-Jiménez[7], Peter Convey[8] & Gonçalo Vieira[9]

Iron supplied by glacial weathering results in pronounced hotspots of biological production in an otherwise iron-limited Southern Ocean Ecosystem. However, glacial iron inputs are thought to be dominated by icebergs. Here we show that surface runoff from three island groups of the maritime Antarctic exports more filterable ($<0.45\,\mu m$) iron ($6–81\,kg\,km^{-2}\,a^{-1}$) than icebergs ($0.0–1.2\,kg\,km^{-2}\,a^{-1}$). Glacier-fed streams also export more acid-soluble iron ($27.0–18,500\,kg\,km^{-2}\,a^{-1}$) associated with suspended sediment than icebergs ($0–241\,kg\,km^{-2}\,a^{-1}$). Significant fluxes of filterable and sediment-derived iron ($1–10\,Gg\,a^{-1}$ and $100–1,000\,Gg\,a^{-1}$, respectively) are therefore likely to be delivered by runoff from the Antarctic continent. Although estuarine removal processes will greatly reduce their availability to coastal ecosystems, our results clearly indicate that riverine iron fluxes need to be accounted for as the volume of Antarctic melt increases in response to 21st century climate change.

[1] Department of Geography, University of Sheffield, Sheffield S10 2TN, UK. [2] Arctic Geology, University Centre in Svalbard (UNIS), N-9171 Longyearbyen, Norway. [3] School of Geographical Sciences, University of Bristol, Bristol BS8 1SS, UK. [4] Department of Life Sciences, Natural History Museum, London SW7 5BD, UK. [5] Departamento de Matematica Aplicada, ETSI de Telecomunicacion, Universidad Politecnica de Madrid, ES-28040 Madrid, Spain. [6] Faculty of Health and Life Sciences, University of Northumbria, Newcastle upon Tyne NE1 8ST, UK. [7] Arctic Biology, University Centre in Svalbard (UNIS), N-9171 Longyearbyen, Norway. [8] British Antarctic Survey, High Cross, Madingley Road, Cambridge CB3 0ET, UK. [9] Centre for Geographical Studies-IGOT, Universidade de Lisboa, Lisbon 1649-003, Portugal. Correspondence and requests for materials should be addressed to A.H. (email: a.j.hodson@sheffield.ac.uk).

Glaciers supply iron to some of Earth's most enigmatic and commercially important marine ecosystems[1]. In Antarctica, icebergs are thought to dominate glacial iron inputs and little attention has been given to runoff[2–4]. However, the volume and extent of surface snow and glacier melt in the Antarctic Peninsula region have increased markedly in response to climate warming over the last half century[5,6]. According to degree-day modelling, meltwater production increased from $28 \pm 12\, Gt\, a^{-1}$ in 1950 to $54 \pm 26\, Gt\, a^{-1}$ by 2000 (ref. 6). However, the proportion of this melt that reaches the sea is uncertain, and the potential contribution of surface streams as transport vectors of bioavailable iron to Antarctic coastal waters is unknown. At present, only subglacial melting is thought to promote iron export to marine ecosystems[7], as supported by direct evidence of enhanced Fe abundance and primary production in the Pine Island Bay polynya south-west of the Antarctic Peninsula[8]. However, subglacial melting is far less responsive to climate warming than surface melting, because it is driven by relatively constant geothermal heating and strain heating as a result of ice flow. Iron transport induced by rapidly changing surface meltwater production therefore needs consideration, especially in the warmer sub-Antarctic and maritime Antarctic islands to the north and west of the Antarctic Peninsua[9–11] (Fig. 1). Here ,there are productive phytoplankton blooms whose response to meltwater runoff may be detected up to 100 km offshore during warm summers[12]. The intense phytoplankton bloom in the South Georgia region has also been found to be more responsive to shallow ( < 20 m) coastal inputs of iron than to dust, and modelling experiments indicate that an assessment of riverine inputs is required to better predict the iron fertilization effects of the island[13]. Consequently, characterizing the iron geochemistry of maritime Antarctic runoff, including both Antarctic and sub-Antarctic islands, could significantly change our understanding of glacier-ocean ecosystem interactions. The glaciers on these islands are also experiencing an earlier and faster response to warming than the large ice masses upon the Antarctic Peninsula and continent; thus representing a basis to understand the future contribution of Antarctic continent melting to the iron budget of the Southern Ocean following projected climate trends[14].

Due to low iron solubility, inputs via colloids and nano-particulates are vital for sustaining the distinct iron fertilization effects seen near islands[2,9,11]. The spatially discrete influence of the islands is also limited by rapid and efficient removal processes near the coast due to scavenging and precipitation[15]. Icebergs may also act as small islands and produce their own local fertilization effects, with less influence from removal processes once they have drifted offshore[3,16,17]. However, unlike runoff, the composition of icebergs is relatively unaffected by rock-water interaction, which has the capacity to enhance the abundance of these important colloids and nano-particulates (also referred to as 'dissolved iron' or 'DFe', on account of their capacity to pass through a 0.45 µm filter[18,19]). In addition, iron can also be extracted from larger suspended sediment particles (SSFe) derived from both runoff and icebergs after a range of abiotic and biotic processes have taken place[18,19]. We therefore present new data on both DFe and SSFe concentrations in surface runoff from sites representative of important islands surrounding the Scotia Sea ecosystem of maritime Antarctica, namely South Georgia, Signy Island (South Orkney Islands) and Livingston Island (South Shetland Islands; Fig. 1). Spring water samples are used to identify the controls upon iron acquisition from sediments following rock-water interaction and before oxidation in the stream system, while stream samples are used to understand iron transport to the sea by runoff and assess its potential importance at regional to continental scales. Iron concentrations were measured using inductively coupled plasma mass spectrometry analysis of samples filtered through 0.45 µm membranes either immediately (DFe), or after 2 months of storage. The difference between these quantities was used to estimate SSFe, which represents the acid-soluble fraction that may be leached from suspended sediments in oxygenated environments. Therefore, samples for DFe were acidified to pH 1.7 immediately after filtration, whilst the samples filtered after 2 months of storage were acidified to pH 1.7 immediately in the field.

Our results represent a multi-site inventory of Antarctic data and allow us to quantify the potential for iron fertilization by runoff in an area where it arguably matter most: the iron-limited Southern Ocean. Temporal and spatial changes in iron transport by snowmelt-dominated runoff are described, showing that the riverine iron flux delivered to coastal waters in the Antarctic Peninsula region is extremely sensitive to climate warming. This is because both the meltwater volume and its iron concentration increase with temperature, resulting in marked increases in the meltwater iron flux, especially in areas where the rate of melting is currently low and dominated by snowmelt. Since this situation is unlike that seen in the more accessible Arctic, where glacier ice ablation heavily subsidises or even exceeds snowmelt in the production of runoff, observations from the Arctic should not be uncritically extrapolated to the Antarctic. Instead, more attention needs to be given the fate of iron delivered to coastal waters by a climate-sensitive Antarctic meltwater flux.

## Results

**Iron acquisition inferred from spring water chemistry.** Spring waters were classified according to the type of sedimentary environment they were sampled from (floodplain sediments, glacial till, soil or talus). Figure 2a shows how variation in the DFe concentrations of individual springs results in a range of almost five orders of magnitude across these categories, from close to $1\, \mu g\, l^{-1}$ in the case of talus springs, through intermediate levels in soils, to levels in excess of $1,000\, \mu g\, l^{-1}$ in springs from glacial tills and glacial floodplains. The highest DFe levels were associated with vegetated wetlands established upon these sediments in South Georgia (up to $36,400\, \mu g\, l^{-1}$). The concentrations here were even greater than in groundwater springs draining acidic (pyritic) rocks with visible iron mineralization (Fe-oxyhydroxide and Schwertmannite) upon King George and Vega Islands of the South Shetland and James Ross Island groups respectively[20,21].

Figure 2a also shows that dissolved organic carbon (DOC) exerts a first-order control upon DFe, but within two distinct groups: first a low-DFe correlation with DOC ($P < 0.05$) that includes soil and talus springs, and second a high-DFe correlation ($P < 0.02$) that includes all springs from glacial tills and floodplain sediments. A notable feature of the high-DFe cluster was their $O_2$ depletion (1.4–88% saturation: Supplementary Data 1, Supplementary Table 1), resulting in a significant ($P < 0.02$, $n = 13$), negative exponential relationship between the DFe concentration and $O_2$ saturation. The process most likely to account for this relationship was sulfide oxidation, a well-known, microbially mediated weathering mechanism in glacial sediments and already described at Signy, King George and Vega islands[20–22]. In addition, iron reduction was detected in the springs with DFe concentrations $> 1,000\, \mu g\, l^{-1}$. Consistent with this, anoxia was found to occur immediately below the water table and we observed the appearance of visible $Fe^{3+}$ flocs due to oxidation of $Fe^{2+}$ in our unacidified samples.

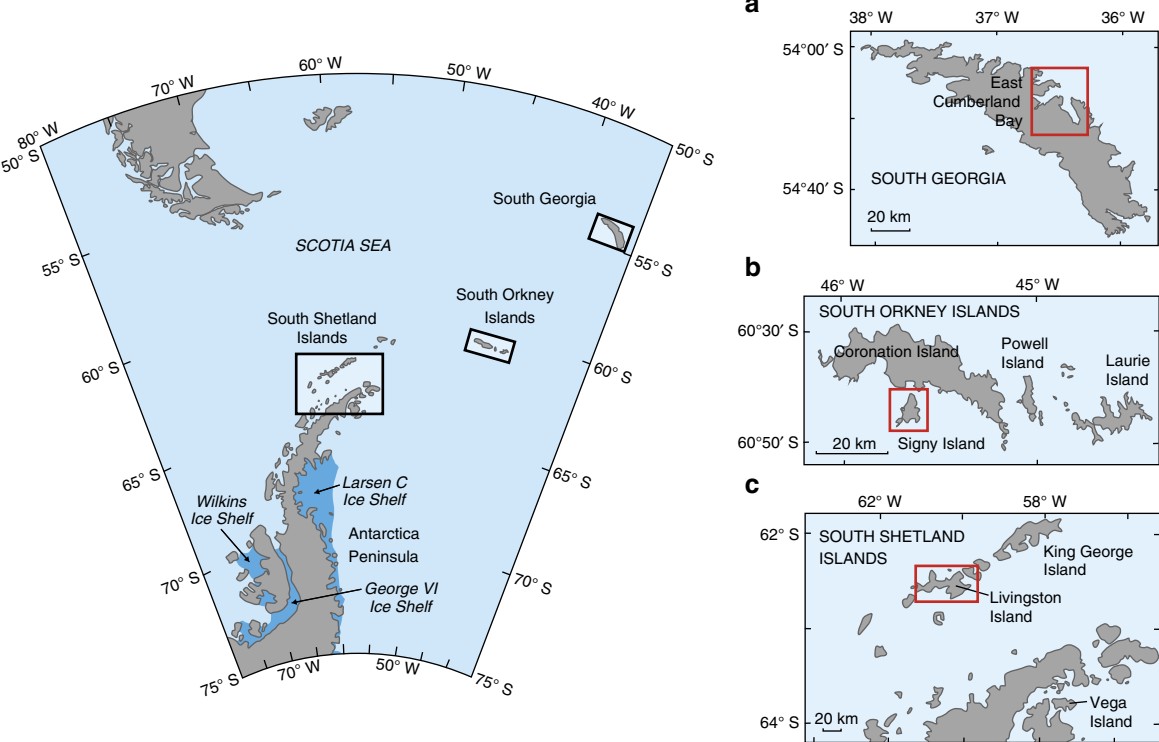

**Figure 1 | The Antarctic Peninsula and the islands visited during this study.** Red boxes show the specific field study areas: (**a**) East Cumberland Bay, South Georgia; (**b**) Signy Island and (**c**) Hurd Peninsula, Livingston Island.

**Stream water iron concentrations.** Figure 2b shows variations in the DFe and SSFe concentrations of the individual streams entering the sea, using average data for sites where multiple samples were collected. The DFe concentrations varied across four orders of magnitude, demonstrating marked site-to-site variability during our study (see also Supplementary Data 1, Supplementary Table 2). Unlike the springs, there was no statistically significant relationship between DFe and DOC, perhaps indicating their decoupling via photolytic destruction of DOC, assimilation or precipitation. The range of DFe concentrations was greatest in non-glacial streams (that is, draining non-glacierized or 'fellfield' catchments) due to low values in steep, soil or talus-covered catchments, and high values in streams fed by the wetlands described above. In contrast, the range of SSFe concentrations was greatest in the streams draining glacierized watersheds (Fig. 2b), and concentrations exceeded those of DFe in all but two cases. The highest SSFe concentrations were associated with turbid glacial meltwaters, especially those on South Georgia, indicating a similar influence of glacierization upon the export of potentially labile iron to that described in the Arctic[18,19].

**Seasonal melt and iron dynamics in streamwater.** Seasonal dynamics of melt, DFe and SSFe were explored in a glacierized (complete glacier ice cover), a fellfield (no glacier ice cover) and a lake-influenced (60% glacier cover and 10% lake cover) catchment upon Signy Island during 2012/13. The summer was notable for low melt rates and intermittent runoff generation, similar to the colder parts of the Antarctic Peninsula region where snow, rather than glacier ice, is the main source of meltwater. Figure 2c shows the relationship between DFe and DOC for all individual samples collected at these different sites, indicating that the relationship between DFe and DOC was only

apparent in the fellfield stream. Here, concentrations of DFe exceeded those observed in the glacierized and lake-influenced catchments. The temporal variations of these data are shown in Fig. 2d–f, which indicate that high DFe concentrations at all three sites coincided with high daily melt rates, resulting in a significant linear correlation ($P < 0.05$) indicative of DFe removal through flushing. Flushing was less apparent with the SSFe data from these sites (Supplementary Data 1, Supplementary Table 3), although a significant linear correlation ($P < 0.05$) between the daily melt rate and SSFe was found at the glacierized site. Therefore, DFe export shows a marked response to short-term warming, because both the concentration and the runoff volume increase. For SSFe, the short-term response to warming is most pronounced where active glaciers are present in the catchment.

**Surface runoff iron export to coastal waters.** Concentrations of DFe and SSFe on the islands were combined with glacier mass balance and precipitation data to estimate rates of riverine export into East Cumberland Bay (South Georgia) and coastal waters surrounding both Hurd Peninsula (Livingston Island) and Signy Island. We also estimated iceberg export from the same catchments using observed calving rates and the average DFe and SSFe contents of Antarctic iceberg samples[4,23]. Table 1 shows the fluxes per unit area (that is, yields), indicating that runoff is a far more significant contributor to coastal DFe supply from land than icebergs from these maritime Antarctic islands. This is largely because surface melting is significant here, and the rock-water interactions identified above enhance the DFe concentration of stream waters to levels more than 20 times greater than average iceberg ice (Table 1). Therefore, icebergs deliver only 2% of the DFe flux into East Cumberland Bay, 0% at Signy (due to recent glacier retreat)

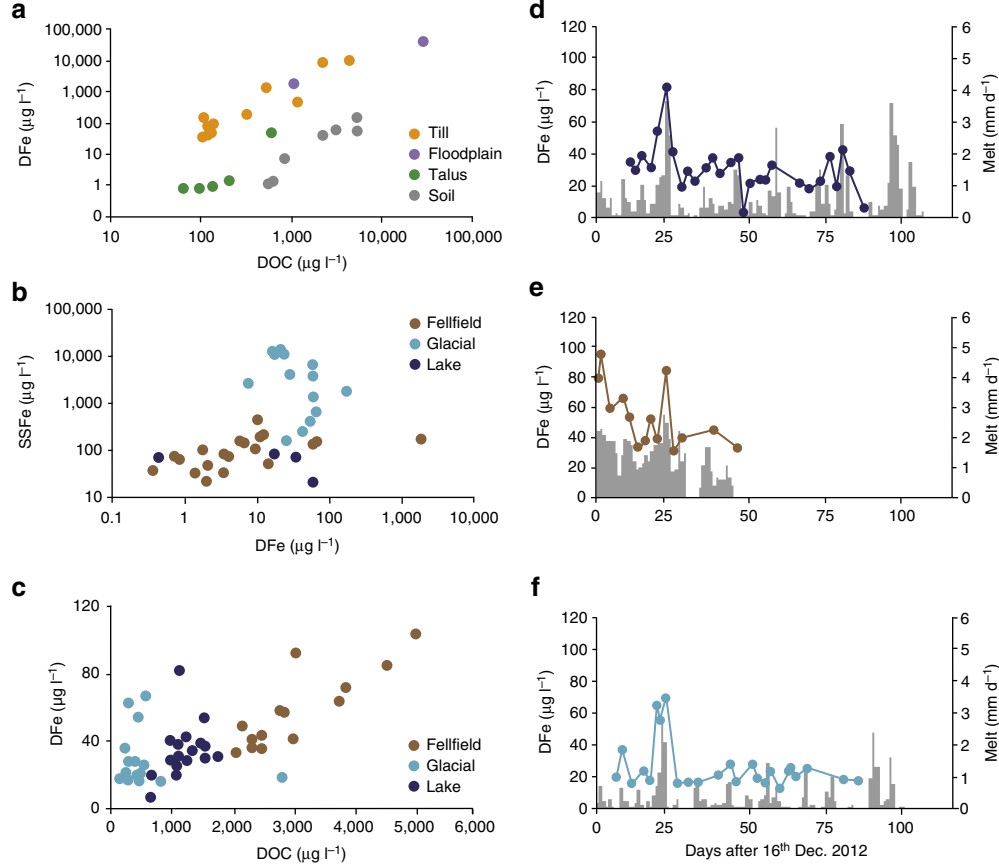

**Figure 2 | DFe relationships with key parameters and seasonal DFe dynamics.** Data points in **a** and **b** represent individual spring and stream sites, respectively (averaged in the case of multiple samples). Data points in **c**–**f** represent individual samples collected upon Signy Island. Colour-coding in **a** distinguishes different sedimentary environments, whilst in **b**–**f**, colours represent different catchment land cover characteristics (fellfield: brown; glacierized: light blue and lake-influenced: dark blue). Figures (**d**–**f**) also show daily melt variations in the lake-influenced, fellfield and glacial catchments upon Signy Island, respectively, to help depict seasonal changes.

**Table 1 | DFe and SSFe concentrations and annual yields for runoff and icebergs.**

| | Livingston Island (Hurd Peninsula) | | | Signy Island | | | South Georgia (East Cumberland Bay) | | |
|---|---|---|---|---|---|---|---|---|---|
| | Glacial runoff | Non-glacial runoff | Icebergs | Glacial runoff | Non-glacial runoff | Icebergs | Glacial runoff | Non-glacial runoff | Icebergs |
| *Concentrations* | | | | | | | | | |
| DFe ($\mu g\,l^{-1}$) | 75.2 ± 62.8 | 32.5 ± 14.7 | 1.84 ± 1.17* | 28.3 ± 14.6 | 57.8 ± 22.4 | – | 28.4 ± 16.2 | 16.3 ± 31.0 | 1.84 ± 1.17* |
| SSFe ($\mu g\,l^{-1}$) | 2,010 ± 1,350 | 586 ± 312 | 380 (150–970)** | 148 ± 310 | 165 ± 241 | – | 13,100 ± 7,980 | 127 ± 135 | 380 (150–970)** |
| *Yields* | | | | | | | | | |
| Water ($m\,a^{-1}$) | 1.08 | 0.01 | 0.25 | 0.10 | 0.10 | 0.00 | 1.50 | 0.59 | 0.64 |
| DFe ($kg\,km^{-2}\,a^{-1}$) | 80.9 | 0.25 | 0.46 | 2.69 | 3.58 | 0.00 | 40.2 | 9.65 | 1.17 |
| SSFe ($kg\,km^{-2}\,a^{-1}$) | 2,160 | 4.59 | 94.8 | 10.3 | 16.5 | 0.00 | 18,450 | 74.8 | 241 |

Concentration values are averages with s.d. for each site (*,** average values for DFe and SSFe, respectively, derived from other studies[4,23]).

and 19% at Livingston Island. The iceberg SSFe fluxes were 4, 0 and 1% of total export at East Cumberland bay, Signy Island and Hurd peninsula, respectively, although SSFe concentrations for icebergs are very poorly constrained, making these estimates very uncertain[4]. The SSFe fluxes in runoff were between 4 and 27 times greater than their corresponding DFe fluxes, with the exception of South Georgia, where the SSFe flux in runoff from glacierized watersheds was 460 times the DFe flux (Table 1). Therefore, runoff must be treated as an important source of both DFe and SSFe to coastal waters in all these environments, and there is clear evidence of SSFe enhancement of the fluxes by glacial erosion.

## Discussion

The DFe concentrations and chemical conditions of spring waters indicated the importance of sulfide oxidation and iron reduction occurring in glacial sediments. The highest DFe concentrations (that is, $>1,000\,\mu g\,l^{-1}$) were associated with the highest DOC concentrations, all of which were samples discharging from vegetated wetlands and floodplains in South Georgia. Therefore, the biogeochemical cycling of iron, especially iron reduction, are enhanced by organic matter accumulation in glacial sediments here (but not on Livingston or Signy Island, where there is much less vegetation). Furthermore, the correlation between DFe and DOC in spring waters suggests that

**Table 2 | Regional estimates of water and iron export to Antarctic coastal waters.**

|  | Low estimate | High estimate |
|---|---|---|
| *Antarctic peninsula (km$^3$a$^{-1}$ for runoff; Gg a$^{-1}$ for iron)* |  |  |
| Surface runoff[6] | 2.90 | 20.0 |
| Surface runoff Dfe | 0.16 | 1.09 |
| Surface runoff SSFe | 17.6 | 122 |
| *Entire Antarctica (km$^3$a$^{-1}$ for runoff or ice; Gg a$^{-1}$ for iron)* |  |  |
| Surface melt runoff | 22.7 | 156.3 |
| Subglacial melt runoff | 32.5 | 97.5 |
| Iceberg water flux[4] | 1177 | 1465 |
| Surface runoff Dfe | 1.24 | 8.52 |
| Surface runoff SSFe | 138 | 950 |
| Subglacial Dfe | 1.77 | 5.32 |
| Subglacial SSFe | 198 | 592 |
| Iceberg DFe[4,23] | 0.79 | 4.41 |
| Iceberg SSFe[4] | 179 | 1,400 |
| Runoff Dfe from ice-free regions[27] | 0.01 | 0.23 |
| Groundwater runoff Dfe[20] | 3.70 | – |

Values for the Antarctic Peninsula use runoff estimates for the year 2000 (ref. 6), whilst values for entire Antarctica use average melt estimates from 1991–2000 for surface runoff[35] and theoretical model predictions for basal melt[36]. Iceberg fluxes use published values not produced by this study[4,23]. Ranges were calculated using upper and lower estimates of the water flux for surface runoff and subglacial runoff. For iceberg fluxes, the ranges also account for the great uncertainty in concentrations (using average concentration ±1 s.d.).

DFe-organic matter complexation occurred: a process that is known to play a significant role in Fe bioavailability after runoff enters coastal waters[15,24]. However, until now, no organic carbon and DFe inter-relationships have been identified in meltwater runoff from the polar regions, suggesting that further characterization of the binding and their effects upon bioavailability of glacial DFe is now required. This is because while binding to humic substances might promote rapid flocculation in sea water, lower molecular weight organic substances can have the opposite effect and promote DFe bioavailability[24].

The estimates of DFe and SSFe yields in Table 1 indicated that surface runoff exports more iron than icebergs, and therefore represents a major contributor of iron into coastal ecosystems within the Scotia Sea. As a consequence, combination of runoff estimates for the entire Antarctic Peninsula[6] with our median DFe and SSFe concentrations for glacial streams (54.5 and 6,080 µg l$^{-1}$ respectively) indicate significant DFe fluxes (0.16–1.09 Gg a$^{-1}$) and SSFe fluxes (17.6–222 Gg a$^{-1}$) at the regional scale. Comparing the efficacy of glacial runoff and iceberg iron transfer from the entire Antarctic continent is fraught with greater uncertainty than the above calculations, but necessitated by there being markedly less surface melting outside the Antarctic Peninsula region and a greater proportion of ice shelves, which will produce large iceberg and subglacial meltwater fluxes[7]. For surface runoff, there are almost no data to describe the proportion of surface melting that manages to reach the sea, whilst for icebergs, there are insufficient data to describe the sediment content of the ice[4]. However, a plausible case for an important flux of iron resulting from coastal melting and surface runoff around the Antarctic continent can still be made if the same proportion of meltwater reaches the sea as has been suggested for the Antarctic Peninsula region[6]. With this being the case, Table 2 shows that DFe fluxes associated with both surface runoff and subglacial runoff lie in the same 1–10 Gg a$^{-1}$ range as icebergs. Similarly, SSFe fluxes lie in the 100–1,000 Gg a$^{-1}$ range, although the upper estimate for icebergs is significantly greater at 1,400 Gg a$^{-1}$. Therefore, greater research effort into the fate of all surface melting in coastal Antarctica and the iron it transports into coastal waters are required if the full response of the coupled Antarctic cryosphere and biosphere to climate change is to be understood. Of great importance is also establishing the impact of estuarine type removal processes, which might account for as much as 90% of the combined runoff and iceberg-derived iron inputs to fjords and shallow coastal waters[15], but will be markedly different for DFe and SSFe in terms of the efficiency and spatial distribution of their removal.

Surface runoff inputs of iron are far more likely to respond to climate warming than subglacial melting and iceberg fluxes. However, since there is evidence of flushing effects in our seasonal data set from Signy Island, a simple, linear response of iron export to the sea following climate warming seems unlikely. Furthermore, combination of our DFe yields with published DFe yields from Antarctic[25] and Arctic[19,26,27] watersheds (Supplementary Data 1, Supplementary Table 4) shows a non-linear response to increasing specific annual runoff (Fig. 3). Therefore, DFe export per unit area increases by nearly three orders of magnitude when the runoff typical of cold, continental Antarctica (in this case the McMurdo Dry Valleys[25]) is increased by an order of magnitude, to the levels observed on Livingston Island. Furthermore, surface energy balance models[28,29] show that an increase of 0.5 °C in air temperature, equivalent to half the increase in summer mean annual air temperature experienced between 1950 and 2000 (ref. 6), enhances glacial runoff by 11, 56 and 34% at Cumberland Bay, Hurd Peninsula and Signy Island, respectively. The greater sensitivity observed at Hurd Peninsula and Signy Island is caused by average summer air temperatures lying close to 0 °C and results in a 73 and a 40% increase in DFe export (yield) respectively, according to the non-linear relationship shown in Fig. 3. The sensitivity of runoff production to climate warming is also consistent with the exponential increase in melt (expressed as the sum of positive degree days) that has resulted from rising air temperatures on both sides of the Antarctic Peninsula in recent decades[6,14]. Since Antarctic melting is projected to double over the interval 2000–2050 (refs 14,30), our results clearly show that a significant increase in the supply of labile iron via surface runoff can be expected in coastal waters, not only from the islands of maritime Antarctica, but also from the Antarctic Peninsula and other parts of the continental land mass.

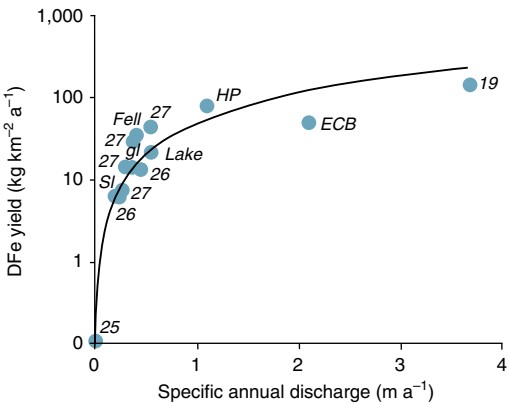

**Figure 3 | Annual runoff and DFe yields from Arctic and Antarctic catchments.** Hurd Peninsula (HP), East Cumberland Bay (ECB) and Signy Island (SI) data are from Table 1. The separate glacial ('gl'), fellfield ('fell') and lake-influenced ('lake') catchments on Signy Island are also shown, whilst numeric labels correspond to published studies in the reference list.

## Methods

**Sampling sites.** New data describing the iron chemistry of runoff upon South Georgia and islands of the South Orkney and South Shetland island groups (Signy Island and Livingston Island, respectively) are presented in this paper. Work was conducted upon Signy Island during a full summer ablation season (November–March) at three sites during the 2012/13 summer. On South Georgia and Livingston Island, sampling was conducted over 6 weeks between mid-January and late February, during 2013 and 2014, respectively. Here, sampling was conducted at a multitude of different sites when runoff fluxes were potentially at their maximum (due to high streamflow).

**Field sampling and analytical methods.** All samples for DFe analysis were syringe-filtered immediately in the field through 0.45 μm filters into pre-cleaned 15 ml eppendorf tubes, before acidification to pH ∼ 1.7 using reagent grade $HNO_3^-$ (AnalaR 65% Normapur, VWR, IL, USA). Pre-cleaning involved washing an initially sterile tube with 10% HCl overnight, before rinsing twice with 18 Mega-ohm deionised water and three times with a small (1–2 ml) aliquot of sample filtrate (DFe) or unfiltered sample (SSFe). Analysis of Fe was then conducted using inductively coupled plasma mass spectrometry (PerkinElmer Elan DRC II, MA, USA). SSFe determination involved 0.45 μm filtration immediately before analysis, after acidification and 2 months of storage at 4 °C in the dark. Although more complicated techniques have been developed to estimate the labile fraction of sediment-bound iron (such as extraction by ascorbic acid[4]), our protocol represents the simplest operationally-defined estimate of the iron that may be leached from suspended sediment. It is therefore realised that in many cases, further processing of this iron might be necessary before assimilation by phytoplankton or benthic algae takes place.

DOC was determined on samples filtered using 0.7 μm glass fibre filters (Whatman GFF), stored in pre-rinsed glass vials and analysed using the membrane conductometric method on a Sievers 5310 Analyser. Precision errors for all the iron and DOC analyses were < 5% according to repeat analyses of mid-range standards, whilst the detection limits were 1.0 μg l$^{-1}$. No contaminants were detected above this limit in the analyses of blank deionised water samples.

At the sampling site, pH, temperature and dissolved oxygen were recorded using Hach Lange HQ 30D meters and dedicated electrodes/sensors. These were calibrated daily before use with the exception of the dissolved $O_2$ measurement, which was conducted using the luminescence method and thus used a factory calibrated sensor tip.

**Flux calculations.** Table 1 and Fig. 3 present site-specific yields (that is, fluxes divided by contributing area) of DFe and SSFe for three large watersheds: East Cumberland Bay (South Georgia: 306 km$^2$, 72% glacierized); Signy Island (South Orkney Islands: 19 km$^2$, 39% glacierized) and Hurd Peninsula (Livingston Island: 73 km$^2$, 99% glacierized). Here we conducted all of our sampling and were also able to use archived data resources to independently estimate the total annual specific runoff of these regions (Q in m a$^{-1}$) according to equation (1):

$$Q = B_s + S^{non-gl} + P^{DJFM} \qquad (1)$$

$B_s$ is the total summer mass loss from glaciers in the area according to stake measurements, $S^{non-gl}$ is the runoff contribution from winter snow accumulation upon non-glacier covered land and $P^{DJFM}$ is the contribution from precipitation

during summer. Here, 'summer' and 'winter' are defined according to the dates used for measurements of accumulation and ablation at stake networks used for Bs estimation (typically 15th December–31st March). All $P^{DJFM}$ was assumed to contribute to runoff. $B_s$ was estimated from the authors' own unpublished observations at Signy Island (see below) and average annual data for Hurd Peninsula and East Cumberland Bay glaciers that is archived by the World Glacier Monitoring Service[31], including data for Hurd Peninsula glaciers from the authors[32]. Glaciers without mass balance data were assumed to have the same change in $B_s$ with elevation as their nearest monitoring site. The mass balance data were applied to the elevation range of glacier ice as specified in 20 m ranges for 2012 by the Inventory of Mountain Glaciers and Ice Caps for the Antarctic Periphery[33]. $S^{non-gl}$ and $P^{DJFM}$ were also expected to increase with elevation, but since there are no published precipitation gradients from any non-glacial sites in Antarctica (to the best of our knowledge), we used precipitation gradients deduced from winter snow accumulation during glacier mass balance monitoring[31]. These gradients were then applied to precipitation records from Juan Carlos I (Spanish) Research Base at Livingston Island and at British Antarctic Survey stations upon Signy Island and Grytviken, South Georgia. Our simplified water balance therefore ignores evaporation, a potential sink for $P^{DJFM}$, largely due to high uncertainty (no data) but also because net condensation is assumed to occur over snow and ice surfaces. Evaporation is potentially significant in low altitude soils on South Georgia, but their areal extent is far less than that of snow, ice and bare rock or moraine. We also assume that the contribution of permafrost thaw to runoff is minimal, on account of the absence of permafrost sediments at South Georgia, and a very low areal extent of permafrost sediments being exposed to thaw at Signy Island and Livingston Island.

DFe and SSFe yields were calculated from the product of Q and the corresponding average iron concentration. The average values were deduced from stream data and from any springs discharging directly into the sea that were not accounted for by the streams. As a result, springs were only considered at Signy and Livingston islands, where colder conditions prevent the development of streams and rivers. The iceberg DFe and SSFe yields were estimated in the same way, using the product of iceberg discharge derived from monitoring[29,34] and the most recent assessment of the average DFe and SSFe contents of Antarctic glacier ice[4,23].

In addition to the larger watersheds described above and presented in Table 1, Fig. 3 shows runoff and DFe yields for the three separate stream catchments upon Signy Island (see also Fig. 2d–f). These were estimated using the same procedure as described above. In so doing, since there are no mass balance data available from the World Glacier Monitoring Service[31], Bs was estimated by the authors from measurements conducted at 18 wooden ablation stakes covering 25% of the glacierized area of the island (that is, the glacial site and the lake site).

We estimated the regional fluxes (that is, annual mass transfers not divided by contributing area) shown in Table 2 from the product of surface meltwater runoff and the median DFe and SSFe concentrations for glacial streams sampled during our study (54.5 and 6,080 μg l$^{-1}$ respectively). Since the Antarctic Peninsula is a very mountainous region, the best available estimates of surface melting and runoff production were derived from a high resolution (1 km grid) degree-day model[6]. This work also accounted for the fraction of surface melting retained by re-freezing: although significant uncertainty in this process results in a wide range of probable runoff values, namely 2.9–20 Gt a$^{-1}$ for the year 2000. For the entire Antarctic melt zone, a high resolution melt flux data set is also available[35] but the proportion of this melt that reaches the sea is not known. Therefore, since most of the observed melt is coastal in origin, we used the same lower and upper limits of melt conversion to runoff as defined by the Antarctic Peninsula study[6]. For the subglacial runoff, the water flux used was defined using an ice sheet thermodynamics model[36]. Other studies have also used this water flux to estimate of basal runoff[7,19], but lack DFe and SSFe concentration data from Antarctica and so are not reported here. Iceberg fluxes used water and SSFe fluxes from the most recent assessment[4]. This does not present DFe concentrations, and so we used the same Antarctic data resource[23] as Table 1 instead.

**Data availability.** All the data used for the calculation of the above fluxes are available from the authors or from the data sets cited above. In addition, data describing the springs and the streams, including their precise location, are presented in Supplementary Data 1 (Supplementary Tables 1 and 2 respectively). Seasonal changes in daily melt and the concentrations of DFe and SSFe in the three Signy Island catchments are found in Supplementary Data 1, Supplementary Table 3. Supplementary Data 1, Supplementary Table 4 contains the DFe yields and specific annual runoff data used for Fig. 3.

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

## Acknowledgements

We acknowledge Natural Environment Research Council grant NE/H014446/1 (A.H., M.S., D.P. and P.C.), Plan Nacional de I + D + I grant CTM2014-56473-R (F.N.) and Fundação para a Ciência e a Tecnologia grant PTDC/AAG-GLO/3908/2012 (G.V.). The Portuguese Polar Program, Bulgarian Antarctic Institute and British Antarctic Survey provided logistical support for the fieldwork. In addition, work at South Georgia was supported by staff at King Edward Point and a Government of South Georgia and South Sandwich Islands Research Grant (A.J., A.H. and A.N.). We thank Steve Colwell and Roger Worland (British Antarctic Survey) for meteorological data from South Georgia and Signy Island.

## Author contributions

A.H., A.N. and M.S. collected the samples, with assistance from Jungblut on South Georgia. The laboratory data were compiled and analysed by A.H., who wrote the manuscript, with equal editorial input from the remaining authors. F.N. provided unpublished meteorological and glaciological data from Livingston Island.

## Additional information

**Competing financial interests:** The authors declare no competing financial interests.

**Publisher's note**: 

