## [Peer Review File · Nature Communications]

Review of Hodson et al by Raiswell

Reviewer #1 (Remarks to the Author):

The authors have dealt well with the main points that I made in an earlier review; specifically with regard to regional estimates and potential removal through an estuarine gradients. I rather think that estuarine removal will still be high, as the Schroth paper concludes, but I am prepared to accept their reluctance to deal with any post-delivery removal, especially as their data shows high concentrations of DOC that might partially mitigate removal from meltwaters derived from vegetated islands. The novelty of the paper rests on it being the first such study from Antarctica (although there are comparable studies of meltwaters from the Arctic) and on the recognition of the extent of iron processing and iron reduction in vegetated areas, and the influence that DOC might have on diminishing estuarine removal. This new features justify, to my mind, publication in Nature Comm which will encourage more such studies from Antarctica.

There are just a few minor issues that need to be resolved.

Page 3, line 2. Subglacial inputs are not runoff. Delete 'by runoff.'

Page 3, line 25. Unclear to me why the effect is exaggerated. Replacing exaggerated by 'limited' would make more sense as you are discussing removal.

Page 4, line 8. Replace scavenged by extracted.

Page 4, line 24. OK but somewhere you need to make the point that your measurements of acidsoluble Fe are not the same as my measurements of ascorbic acid soluble.

Page 5, line 9. This seems unclear and apparently contradictory... 'where melting is currently low and dominated by snow melt'. Melting of what else is low-ice? Clarify.

Page 5, line 10. The Arctic is more accessible but is it dominated by glacier-ice melt, as compared to what?

Page 6, lines 15 and 16. Better to say as follows. In addition, anoxia occurs immediately below the water table and iron reduction produces springs with high concentrations of Fe > 1000 microg/litre. Consistent with this we observed the appearance of visible Fe³⁺ flocs due to oxidation of Fe²⁺ in our unacidified samples.

References. The Raiswell Biogeosciences paper has now been published as Biogeosciences vol 13, pages 1-14.

Reviewer #2 (Remarks to the Author):

Review of 'Climatically-sensitive transfer of iron to maritime Antarctic ecosystems by surface runoff' for Nature Communications

This is my second review of this manuscript (Reviewer 2 for the Nature Geosciences version), and I find it much improved from a writing and narrative standpoint. In particular, I appreciate the authors' efforts to clarify the writing and narrative structure. I also appreciate their effort to include some discussion of the estuarine removal process, and I buy their argument for not including a removal coefficient in their comparison to iceberg-derived fluxes. From an iron standpoint, I do think the overly simplistic operational approach used by the authors really limits the novelty of this work in the context of iron biogeochemistry in glacierized systems-but perhaps, as argued by the authors and Dr. Raiswell, because of the global importance of the site coupled

with its remoteness, publication in Nature Communications may be warranted depending on the perspective of the AE after integrating reviewer feedback.

From a general standpoint, developing a schematic type figure that illustrates what we have learned about Fe dynamics and supply to the coastal ocean across the Antarctic landscape could be a powerful tool for readers and certainly promote impact and citation of this particular manuscript-this could also be used to promote further studies that you suggest. Perhaps this has already been developed for presentations or proposals, but I suggest inclusion of something like that in the manuscript if possible. I also suggest replacing 'glacial' with 'glacierized' when discussing watersheds with active glaciation-I have been scolded by glacial hydrologists regarding this terminology in the past. Below are some explicit recommendations for improvement in this version of the text. I hope that you find my feedback useful.

Specific Comments:

Title: seems like you need to mention landscape characteristics and say something related to the scale of delivery in the title as well, as that is much of the focus of your analyses-how about flipping this around to something like 'Extensive delivery of surface runoff-derived iron to the maritime Antarctic driven by climate and landscape characteristics (or provenance or land cover)'. My suggestion is not perfect, but you get the point.

Pg 2

line 25- change 'glacially crushed bedrock' to 'glacial weathering'

line 47 delete 'rather' and 'so'

pg 4 line 90 ICP-MS is the correct acronym, and it needs to be defined somewhere in the text

pg 5, line 112 I don't like the use of the term 'acquisition' here-it seems to me like provenance is a better term for what you are talking about, but correct me if I am wrong-I have not seen acquisition used in this context in the literature, but if you are trying to be consistent with a previous body of work, please disregard.

Pg-6-7 lines 139-165 and Figure 2b,c-I find the presentation of these figures/results a little misleading and some clarification is needed in the text. For example, in figure 2B you are presenting AVERAGE (?) DFe and SSFe concentrations for ALL sites to demonstrate that there is dramatic variability in mean concentrations across this range of environments that you sampled and that there is a predictable difference in Fe partitioning between the glacierized and non glacierized catchments. The second plot is used to demonstrate that there is only a Fe-DOC association that is consistent over time in the fellfield catchment at ONE site, but not at the individual nearby proglacial lake and glacierized stream sites that were also more intensively monitored over time. It is hard to glean this from the current discussion of results. I would start each of these paragraphs (lines 139 and 151) explicitly describing the data, so that differentiation is clear to the reader and the data context is clear. You should also emphasize/clarify this in the caption.

Line 140-142-you are also likely looking at different species of Fe in the DFe fraction-mineral vs. organic colloids-eg more mineral colloid relative to organic in glacial catchments vs highly variable distribution in 'fellfield' sites depending on landcover characteristics-this is widely observed elsewhere in a vast body of more specific size fractionation studies. This same comment applies to your time series from each catchment when talking about Fe-DOC relationships.

Lines 161-163-you show no evidence of 'flushing' for SSFe-I suggest adding this number to the time series with appropriate scaling as to not overprint the DFe data-Alternatively, use supplemental information for the SSFe time series and then reference appropriately in the results.

You are discussing flushing of solutes and sediment, which are not necessarily following the same hydrologic pathways nor coming from the same source-this warrants discussion somewhere.

Pg 8 line 189 Delete 'Data describing'

Pg 9 line 211 and 212-I would combine these paragraphs

pg 10 line 231-probably worth mentioning that the impact of the estuary and resultant removal coefficient will be different for riverine DFe and SSFe, and that the pattern of removal of SSFe in particular would likely vary tremendously in space and based on hydrodynamic conditions in the river and estuary.

Reviewer 1

The authors have dealt well with the main points that I made in an earlier review; specifically with regard to regional estimates and potential removal through an estuarine gradients. I rather think that estuarine removal will still be high, as the Schroth paper concludes, but I am prepared to accept their reluctance to deal with any post-delivery removal, especially as their data shows high concentrations of DOC that might partially mitigate removal from meltwaters derived from vegetated islands. The novelty of the paper rests on it being the first such study from Antarctica (although there are comparable studies of meltwaters from the Arctic) and on the recognition of the extent of iron processing and iron reduction in vegetated areas, and the influence that DOC might have on diminishing estuarine removal. This new features justify, to my mind, publication in Nature Comm which will encourage more such studies from Antarctica.

We appreciate these comments a lot.

There are just a few minor issues that need to be resolved.

Page 3, line 2. Subglacial inputs are not runoff. Delete 'by runoff.'

AJH: DONE

Page 3, line 25. Unclear to me why the effect is exaggerated. Replacing exaggerated by 'limited' would make more sense as you are discussing removal.

AJH: DONE

Page 4, line 8. Replace scavenged by extracted.

AJH: DONE

Page 4, line 24. OK but somewhere you need to make the point that your measurements of acid soluble Fe are not the same as my measurements of ascorbic acid soluble.

AJH: DONE IN METHODS: see end of first paragraph in "Field Sampling and Analytical Methods" section

Page 5, line 9. This seems unclear and apparently contradictory... 'where melting is currently low and dominated by snow melt'. Melting of what else is low-ice? Clarify.

AJH: Done – clarified by stating that glacial environments with low rates of melting tend to remain snow-covered and thus runoff is dominated by snowmelt, rather than melting of the underlying glacier ice (such as in the Arctic). The changes are on p5, lines 8-14.

Page 5, line 10. The Arctic is more accessible but is it dominated by glacier-ice melt, as compared to what?

AJH: see above

Page 6, lines 15 and 16. Better to say as follows. In addition, anoxia occurs immediately below the water table and iron reduction produces springs with high concentrations of Fe > 1000 microg/litre. Consistent with this we observed the appearance of visible Fe³⁺ flocs due to oxidation of Fe²⁺ in our unacidified samples.

AJH: DONE: see p6 line 18 -22.

References. The Raiswell Biogeosciences paper has now been published as Biogeosciences vol 13, pages 1-14.

AJH: The full reference is now inserted

Reviewer #2 (Remarks to the Author):

Review of 'Climatically-sensitive transfer of iron to maritime Antarctic ecosystems by surface runoff' for Nature Communications

This is my second review of this manuscript (Reviewer 2 for the Nature Geosciences version), and I find it much improved from a writing and narrative standpoint. In particular, I appreciate the authors' efforts to clarify the writing and narrative structure. I also appreciate their effort to include some discussion of the estuarine removal process, and I buy their argument for not including a removal coefficient in their comparison to iceberg-derived fluxes. From an iron standpoint, I do think the overly simplistic operational approach used by the authors really limits the novelty of this work in the context of iron biogeochemistry in glacierized systems-but perhaps, as argued by the authors and Dr. Raiswell, because of the global importance of the site coupled with its remoteness, publication in Nature Communications may be warranted depending on the perspective of the AE after integrating reviewer feedback.

From a general standpoint, developing a schematic type figure that illustrates what we have learned about Fe dynamics and supply to the coastal ocean across the Antarctic landscape could be powerful tool for readers and certainly promote impact and citation of this particular manuscript-this could also be used to promote further studies that you suggest. Perhaps this has already been developed for presentations or proposals, but I suggest inclusion of something like that in the manuscript if possible.

AJH: We like this idea a lot but it would require giving much emphasis to subglacial processes that we have not covered in detail here. We are happy to do this, but would rather save it for the synthesis paper that we are writing about the delivery of all inorganic nutrients from glaciers to coastal waters (which mostly includes observations from the Arctic). We therefore welcome your views on this but would prefer to keep the present paper entirely focussed upon Antarctic surface melting.

I also suggest replacing 'glacial' with 'glacierized' when discussing watersheds with active glaciation-I have been scolded by glacial hydrologists regarding this terminology in the past.

AJH: Agreed and done. Where we refer to a watershed, we use "glacierized". Where we refer to a stream, we use "glacial", since "glacierized stream" makes no sense. However, we agree that using "glacial" watershed should be avoided. See p7, lines 8 – 16; p8, lines 2 and 8; p9, line 6.

Below are some explicit recommendations for improvement in this version of the text. I hope that you find my feedback useful.

Title: seems like you need to mention landscape characteristics and say something related to the scale of delivery in the title as well, as that is much of the focus of your analyses-how about flipping this around to something like 'Extensive delivery of surface runoff-derived

iron to the maritime Antarctic driven by climate and landscape characteristics (or provenance or land cover)'. My suggestion is not perfect, but you get the point.

AJH: We liked the suggestion and wanted to append "modified by land cover characteristics" to the original title, but it resulted in 16 words. Therefore we left the title unchanged.

Pg 2

line 25- change 'glacially crushed bedrock' to 'glacial weathering'

line 47 delete 'rather' and 'so'

AJH: Both done

pg 4 line 90 ICP-MS is the correct acronym, and it needs to be defined somewhere in the text

AJH: Done: see p4, line 19 and p 13, line 3.

pg 5, line 112 I don't like the use of the term 'acquisition' here-it seems to me like provenance is a better term for what you are talking about, but correct me if I am wrong-I have not seen acquisition used in this context in the literature, but if you are trying to be consistent with a previous body of work, please disregard.

AJH: we have no strong feelings here either, but to us, "acquisition" means the process and "provenance" defines the source. We are mostly, but not entirely, writing about the processes – so we prefer to stick with "acquisition. For example, while terms like "Sulphide oxidation" reveal both the process and (to some extent) the mineral source – terms like Iron reduction are only about the process. We are not entirely certain about all the sources either, as we have no basis for distinguishing between different Fe-containing minerals in this paper.

Pg-6-7 lines 139-165 and Figure 2b,c-I find the presentation of these figures/results a little misleading and some clarification is needed in the text. For example, in figure 2B you are presenting AVERAGE (?) DFe and SSFe concentrations for ALL sites to demonstrate that there is dramatic variability in mean concentrations across this range of environments that you sampled and that there is a predictable difference in Fe partitioning between the glacierized and non glacierized catchments. The second plot is used to demonstrate that there is only a Fe-DOC association that is consistent over time in the fellfield catchment at ONE site, but not at the individual nearby proglacial lake and glacierized stream sites that were also more intensively monitored over time. It is hard to glean this from the current discussion of results. I would start each of these paragraphs (lines 139 and 151) explicitly describing the data, so that differentiation is clear to the reader and the data context is clear. You should also emphasize/clarify this in the caption.

AJH: Thanks – we have followed this suggestion, beginning with Figure 2a and appreciate the advice on improving the clarity. The changes start on p5, line 21; p7, lines 1 and 23, and p8, line 3. We have also improved the caption to Figure 2.

Line 140-142-you are also likely looking at different species of Fe in the DFe fraction-mineral vs. organic colloids-eg more mineral colloid relative to organic in glacial catchments vs highly variable distribution in 'fellfield' sites depending on landcover characteristics-this is widely observed elsewhere in a vast body of more specific size

fractionation studies. This same comment applies to your time series from each catchment when talking about Fe-DOC relationships.

AJH: Agreed.

Lines 161-163-you show no evidence of 'flushing' for SSFe-I suggest adding this number to the time series with appropriate scaling as to not overprint the DFe data-Alternatively, use supplemental information for the SSFe time series and then reference appropriately in the results. You are discussing flushing of solutes and sediment, which are not necessarily following the same hydrologic pathways nor coming from the same source-this warrants discussion somewhere.

We agree that SSFe and DFe might not be following the same hydrologic pathways and conducted more analysis of the data. This suggested that flushing of SSFe is only a statistically robust finding at the glacierized site and so we have amended our discussion accordingly on p8, lines, 6-12.

Pg 8 line 189 Delete 'Data describing'

AJH: done.

Pg 9 line 211 and 212-I would combine these paragraphs

AJH: Agreed and done.

pg 10 line 231-probably worth mentioning that the impact of the estuary and resultant removal coefficient will be different for riverine DFe and SSFe, and that the pattern of removal of SSFe in particular would likely vary tremendously in space and based on hydrodynamic conditions in the river and estuary.

AJH: A good suggestion and therefore inserted. The changes are on p11, lines, 5-7.